# Pattern of Respiratory Viruses among Pilgrims during 2019 Hajj Season Who Sought Healthcare Due to Severe Respiratory Symptoms

**DOI:** 10.3390/pathogens10030315

**Published:** 2021-03-08

**Authors:** Salma M. Alsayed, Thamir A. Alandijany, Sherif A. El-Kafrawy, Ahmed M. Hassan, Leena H. Bajrai, Arwa A. Faizo, Eman A. Mulla, Lujain S. Aljahdali, Khalid M. Alquthami, Alimuddin Zumla, Esam I. Azhar

**Affiliations:** 1Special Infectious Agents Unit, King Fahd Medical Research Center, King Abdulaziz University, Jeddah 21589, Saudi Arabia; Salsayed0052@stu.kau.edu.sa (S.M.A.); talandijany@kau.edu.sa (T.A.A.); saelkfrawy@kau.edu.sa (S.A.E.-K.); hmsahmed@kau.edu.sa (A.M.H.); lbajrai@kau.edu.sa (L.H.B.); aafaizo@kau.edu.sa (A.A.F.); a.zumla@ucl.ac.uk (A.Z.); 2Department of Medical Laboratory Technology, Faculty of Applied Medical Sciences, King Abdulaziz University, Jeddah 21589, Saudi Arabia; 3Department of Nursing, Faculty of Al-Qunfudah Health Sciences, Umm Al-Qura University, Makkah 28821, Saudi Arabia; 4Department of Biochemistry, Faculty of Sciences, King Abdulaziz University, Jeddah 21589, Saudi Arabia; 5Makkah Regional Lab, Ministry of Health, Makkah 25321, Saudi Arabia; emulla@moh.gov.sa; 6Saudi Center for Disease Prevention and Control, Public Health Lab, Ministry of Health, Riyadh 13354, Saudi Arabia; laljjahdali@moh.gov.sa; 7Al-Noor Specialist Hospital, Ministry of Health, Makkah 24241, Saudi Arabia; kalquthami@moh.gov.sa; 8Division of Infection and Immunity, Centre for Clinical Microbiology, University College London Royal Free Campus, London WC1E 6DE, UK; 9NIHR Biomedical Research Centre, UCL Hospitals NHS Foundation Trust, London W1T 7DN, UK

**Keywords:** Hajj, mass gathering, pilgrims, respiratory tract infection, respiratory symptoms, viruses, carriage, etiology, Saudi Arabia

## Abstract

The aim of our study was to define the spectrum of viral infections in pilgrims with acute respiratory tract illnesses presenting to healthcare facilities around the holy places in Makkah, Saudi Arabia during the 2019 Hajj pilgrimage. During the five days of Hajj, a total of 185 pilgrims were enrolled in the study. Nasopharyngeal swabs (NPSs) of 126/185 patients (68.11%) tested positive for one or more respiratory viruses by PCR. Among the 126 pilgrims whose NPS were PCR positive: (a) there were 93/126 (74%) with a single virus infection, (b) 33/126 (26%) with coinfection with more than one virus (up to four viruses): of these, 25/33 cases had coinfection with two viruses; 6/33 were infected with three viruses, while the remaining 2/33 patients had infection with four viruses. Human rhinovirus (HRV) was the most common detected viruses with 53 cases (42.06%), followed by 27 (21.43%) cases of influenza A (H1N1), and 23 (18.25%) cases of influenza A other than H1N1. Twenty-five cases of CoV-229E (19.84%) were detected more than other coronavirus members (5 CoV-OC43 (3.97%), 4 CoV-HKU1 (3.17%), and 1 CoV-NL63 (0.79%)). PIV-3 was detected in 8 cases (6.35%). A single case (0.79%) of PIV-1 and PIV-4 were found. HMPV represented 5 (3.97%), RSV and influenza B 4 (3.17%) for each, and Parechovirus 1 (0.79%). Enterovirus, Bocavirus, and *M. pneumoniae* were not detected. Whether identification of viral nucleic acid represents nasopharyngeal carriage or specific causal etiology of RTI remains to be defined. Large controlled cohort studies (pre-Hajj, during Hajj, and post-Hajj) are required to define the carriage rates and the specific etiology and causal roles of specific individual viruses or combination of viruses in the pathogenesis of respiratory tract infections in pilgrims participating in the annual Hajj. Studies of the specific microbial etiology of respiratory track infections (RTIs) at mass gathering religious events remain a priority, especially in light of the novel SARS-CoV-2 pandemic.

## 1. Introduction

Viral respiratory tract infections (RTIs) are one of the most important health problems encountered by pilgrims performing the annual Hajj pilgrimage [1,2]. The Hajj is the largest recurring mass gathering religious event where over 2 million pilgrims come from over 184 countries to visit the holy cities of Makkah and Madinah in Saudi Arabia [1,2,3]. The Hajj poses unique challenges to the public health authorities of Saudi Arabia, [4,5] and of the 184 countries from where pilgrims originate and return. The Hajj requires careful logistic planning including infection control strategies to ensure the safety of pilgrims and of local Saudi citizens and global population [6,7]. However, prolonged close contact between pilgrims when performing religious rituals, and overcrowding increases susceptibility of pilgrims to acquisition of airborne microorganisms. Whilst several epidemiological and clinical studies have identified the spectrum of respiratory tract viral infections in cohorts of Hajj pilgrims of varying sizes, it remains important to keep track of the viral etiology at recurring mass gathering religious events.

The aim of our study was to define the spectrum of viral infections in pilgrims with acute respiratory tract illnesses presenting to healthcare facilities around the holy places in Makkah, Saudi Arabia during the 2019 Hajj pilgrimage.

## 2. Materials and Methods

### 2.1. Ethical Approval

This study was approved by the Research Ethics Committee (REC), Unit of Biomedical Ethics, Faculty of Medicine, King Abdulaziz University (KAU) (Reference No 569-20). Written consent forms were obtained from participants.

### 2.2. Study Design

This was a cross-sectional analytical study, which screened for respiratory viruses in the nasopharyngeal swabs of pilgrims with respiratory tract illnesses presenting to healthcare facilities in the holy places during the 2019 Hajj.

Study population and enrolment sites: Pilgrims with severe acute respiratory tract symptoms who presented to, and were initially hospitalized at seven healthcare facilities in Makkah, Arafat, and Mina during the first five days of the 2019 Hajj.

Clinical Specimens and patient information: nasopharyngeal swabs (NS) were collected in viral transport media (VTM) and stored at −80 °C in the Special Infectious Agents Unit (SIAU), King Fahd Medical Research Center (KFMRC). Demographic data including nationality, age, and gender from all participants was also obtained.

### 2.3. RNA Extraction and Molecular Detection of Respiratory Viruses

Nucleic acids were extracted using an ExiPrep™ 96 Viral DNA/RNA Kit through ExiPrep™ 96 Lite (Catalog No. A-5250) (Bioneer, Seoul, Korea) as per the manufacturer’s instructions. Extracted RNA was subjected to multiplex real-time PCR utilizing fast-track respiratory pathogens 21 kit (Fast-track Diagnostics, Luxembourg) that enables the detection of influenza A (H1N1) virus; influenza B (Flu B) viruses; human rhinoviruses (HRV); human coronaviruses (CoV-229E, CoV-NL63, CoV-HKU1, and CoV-OC4); human metapneumoviruses (hMPV) A/B; human parainfluenza viruses (PIV-1, PIV-2, PIV-3, and PIV-4); human bocavirus; human adenovirus; respiratory syncytial virus (RSV) A/B; enterovirus; human parechovirus; and *Mycoplasma pneumoniae*. Amplification was performed using the QuantStudio 12K Flex Real-Time PCR System (Applied biosystem—Thermo Fisher Scientific, Waltham, USA).

### 2.4. Statistical Analysis

The statistical analysis of the data was carried out using the statistical software package SPSS version 25.0 (SPSS Inc., Chicago, IL, USA). All categorical variables (gender and the clinical status of pilgrims) presented as numbers (*n*) and percentages (%). The continuous variable such as age was expressed as mean ± SD. A non-parametric test (Chi-square test) was applied to compare the proportions and determine the significance of association between categorical variables. *p*-values less than 0.05 was considered as statistically significant. Graph drawing was performed by using GraphPad Prism software version 9 (GraphPad Software, La Jolla, CA, USA).

## 3. Results

### 3.1. Characteristics of the Pilgrims Enrolled in the Study

A total of 185 pilgrims with severe acute respiratory symptoms who visited the medical facilities in the Holy Places of Makkah during the five days of the 2019 Hajj season were enrolled in the study. All patients were hospitalized and initially were tested for MERS-CoV. All patients enrolled tested negative for MERS-CoV. Table 1 shows the demographics of the study population. The age of participants ranged from 2 days to 88 years with a mean of 58 ± 14.74 years. More than half (60.54%) of them aged 60 years and older. The total number of male and female were 122 (65.95%) and 63 (34.05%), respectively. Out of the 185 patients, 126 (68.11%) tested positives for one or more respiratory viruses. Males accounted for 81 (64.29%) cases while the remaining 45 (35.71%) were female. Most cases were associated with pilgrims aged 60 years old and older (66.67%).

### 3.2. Detection of Respiratory Viruses by Pligrims’ Country of Origin

Patients participating in this study descended from 37 countries, mainly from Asia (*n* = 119; 64%) and Africa (*n* = 56; 30.3%), followed by Europe (*n* = 5; 2.70%), North America for each (*n* = 4; 2.2%), then Oceania (*n* = 1; 0.54%). Respiratory viruses have been more frequently detected in patients arriving from countries providing annually the highest number of pilgrims [8]. Pilgrims from Indonesia followed by India and Egypt represented the most positive cases of respiratory viruses (more than 10 cases). Five to 10 positive cases reported from pilgrims arrived from Morocco, Saudi Arabia, Bangladesh, Pakistan, Nigeria, Somalia, and Sudan. This is followed by those came from Turkey, Iran, Kazakhstan, and Ethiopia with 1–4 positive cases. No positive cases were identified in pilgrims from China, Algeria, Guiana, France, and Côte d’Ivoire (Figure 1).

### 3.3. The Prevalence of Single Infection and Coinfection among the Pilgrims

Nasopharyngeal samples of 126/185 patients (68.11%) tested positive for one or more respiratory viruses by PCR (Figure 2), while the remaining 59 were negative. Among the 126 pilgrims whose NP swabs were PCR positive: (a) there were 93/126 (74%) with a single virus infection and (b) 33/126 (26%) with coinfection with more than one virus (up to four viruses): Of these 25 cases had coinfection with two viruses; 6 were infected with three viruses, while the remaining 2 patients had infection with four viruses (Figure 2).

Human rhinovirus (HRV) was the most common detected viruses with 53 cases (42.06%), followed by 27 (21.43%) cases of influenza A (H1N1), and 23 (18.25%) cases of influenza A other than H1N1. Twenty-five cases of CoV-229E (19.84%) were detected more than other coronavirus members (5 CoV-OC43 (3.97%), 4 CoV-HKU1 (3.17%), and 1 CoV-NL63 (0.79%)). PIV-3 was detected in 8 cases (6.35%). A single case (0.79%) of PIV-1 and PIV-4 were found. HMPV represented 5 (3.97%), RSV and influenza B 4 (3.17%) for each, and Parechovirus 1 (0.79%). Enterovirus, Bocavirus, and *M. pneumoniae* were not detected (Figure 3). The number of cases for each virus and the frequency (%) according to gender and age group are shown (Table 2).

The most common virus detected in multiple virus coinfection cases was HRV with 28 cases out of 33 (84.85%). HRV was commonly accompanied with CoV-229E (*n* = 6), influenza A (H1N1) and CoV-OC43 (*n* = 3 for each), followed by adenovirus or influenza A not H1N1 (*n* = 2) (Figure 4).

## 4. Discussion

In our study the human rhinovirus (HRV) was the most commonly detected virus with 53 cases (42.06%), followed by 27 (21.43%) cases of influenza A (H1N1), and 23 (18.25%) cases of influenza A other than H1N1. Twenty-five cases of CoV-229E (19.84%) were detected more than other coronavirus members (five CoV-OC43 (3.97%), four CoV-HKU1 (3.17%), and one CoV-NL63 (0.79%)). PIV-3 was detected in eight cases (6.35%). There was one case of PIV-1 and PIV-4 that was found. HMPV represented 5 (3.17%), RSV and Flu B 4 (3.17%) for each, and Parechovirus 1 (0.79%). Enterovirus, Bocavirus, and *M. pneumoniae* were not detected. As previously described [8,9,10,11,12], our study also shows that these respiratory viral infections are common during Hajj. However, the specific viral type, frequency, and combinations of viruses detected vary between studies due to differences in the study design, target population, and other variables [8,9,10,11,12].

As per pilgrims’ home country, most cases associated with those arrived from Indonesia, India, and Egypt (10 or more cases) (Figure 1). Cases of single and coinfection with up to four viruses were identified (Figure 2). Out of the 126 pilgrims with PCR positive results, single infection was detected in 93 (74%) patients. Coinfection was detected in the remaining 33 (26%) cases as follows: 25 patients with two viruses, 6 patients with three viruses, and 2 patients with four viruses (Figure 2). HRV was the most commonly detected virus followed by influenza A (H1N1) then CoV-229E, and influenza A other than H1N1 (Figure 3) (Table 2). Several studies have shown similar trends [8,11,13,14,15,16]. Non-Middle East respiratory syndrome (MERS) HCoVs including CoV-OC43, CoV-HKU1, and CoV-NL63 were readily detected in this study, which also corresponds to previous reports (Figure 3) (Table 2) [8,9,11,17]. Unlike some published reports [8,18,19,20,21,22,23], our data showed most cases were significantly associated with pilgrims over 60 years of age (Table 1). The vulnerability of elders to infections is probably due to a less efficient immune response, which is provoked by factors such as chronic diseases, and physical and mental stress [24,25,26]. This study highlights the need for increasing attention and provision of preventive infection control strategies to this age group.

Respiratory tract viral infections (RTIs) are the most common infections among pilgrims and represent the leading cause of most hospitalizations [7,8,27,28]. In both international and domestic pilgrims [24,29]. Over the last decade, several epidemiological studies have been carried out on pilgrims showing high prevalence of RTIs mostly attributed to respiratory viruses [11,30,31,32,33]. A longitudinal study conducted over 15 years in all related papers published between 2003 and 2018 showed wide variation in the prevalence rates of cough (1.9–91.5%), pneumonia (0.2–54.8%), and influenza-like illness (ILI; 8–78.2%) among pilgrims [30,34]. HRV, influenza viruses, and human coronaviruses other than MERS were the most commonly detected viruses among symptomatic pilgrims, although other viruses like parainfluenza virus, enterovirus, and adenovirus have also been detected [11]. Both single infection and coinfection are common among pilgrims. Indeed, we reported a single infection and coinfection with up to four viruses among pilgrims attended the health care facilities in the holy places in Makkah. Influenza A viruses (Flu A), non-MERS human coronaviruses, HRV, and Influenza B viruses (Flu B) were shown to be the most common viruses identified [8].

Transmission and control of respiratory viral infections are critical health challenges, particularly in a country like Saudi Arabia, which receives annually about 3 million pilgrims and more than 20 million religious tourists from more than 180 countries. Overcrowding of individuals with close contact in small areas provide a favorable environment for transmission of respiratory infections. The public and global health concerns increase during Hajj because of the potential of importing and exporting viruses all over the world through pilgrims [35]. Active surveillance of respiratory viruses during Hajj seasons enables the implementation of directed and efficient prevention and management strategies to control the spread of respiratory viral infections. Among national efforts to ensure the health condition of pilgrims and improving the preventive measures in every Hajj season, this study aimed to identify pattern of respiratory viruses and determine their prevalence among symptomatic pilgrims who visited the healthcare facilities in the holy places during the 2019 Hajj season.

Importantly, our study showed that 26% of the pilgrims with PCR positive results showed a coinfection with more than one virus (up to four viruses). Of these, 25/33 cases had coinfection with two viruses; 6/33 were infected with three viruses, while the remaining 2/33 patients had infection with four viruses. Whether identification of viral nucleic acid represents nasopharyngeal carriage or specific causal etiology of RTI remains to be defined. The PCR test only detects the presence of nucleic acid and cannot determine whether the virus is alive, or it represents bits of nucleic acid from previous infection. The identification of the virus does not mean it is the causative pathogen since nasopharyngeal carriage of certain viruses is common in most populations. Thus, large controlled cohort studies (pre-Hajj, during Hajj, and post-Hajj) are required to define the carriage rates and the specific etiology and causal roles of specific individual viruses or combination of viruses in the pathogenesis of respiratory tract infections in pilgrims participating in the annual Hajj. As Saudi Arabia monitors the global SARS-CoV-2 situation and the numbers of pilgrims allowed to perform the Hajj under strict public health measures of social distancing and wearing of masks, an opportunity to study the effects of stricter infection control measures at mass gathering arises. Studies of the evolution and changes in the viral flora in the nasopharynx of pilgrims and those who develop symptoms of RTI would be appropriate.

In summary, respiratory viruses continue to be a possible public health concern for individuals and nations during Hajj seasons. Herein, we demonstrated the pattern of respiratory viruses among pilgrims during the 2019 Hajj Season. There is a need for larger studies to comprehensively determine the pattern of respiratory viruses among pilgrims. Furthermore, implementation of continuous surveillance is of great importance to accurately estimate the potential risk and impose efficient infection control strategies during this annual mass gathering event. Studies of the specific microbial etiology of RTIs at mass gathering religious events remain a priority, especially in light of the novel SARS-CoV-2 pandemic.

## Figures and Tables

**Figure 1 pathogens-10-00315-f001:**
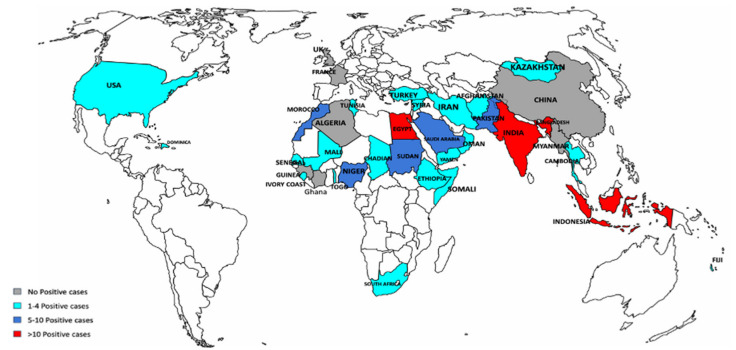
Distribution of positive cases as per pilgrims’ nationalities, Countries color indicate the number of positive cases: grey = no positive cases, light blue = 1–4 cases, blue = 5–10 cases, and red = more than 10 cases.

**Figure 2 pathogens-10-00315-f002:**
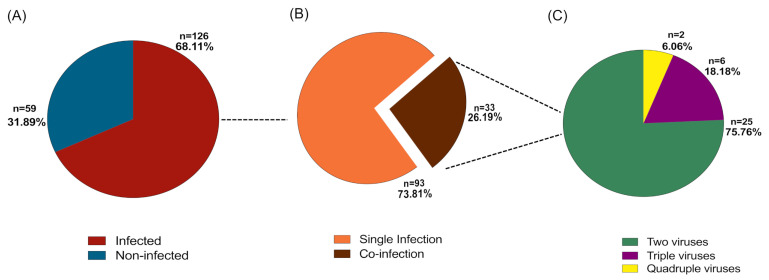
The prevalence of single infection and coinfection among the pilgrims, (**A**) pie chart shows the numbers (*n*) and percentages (%) of infected (red) and non-infected (blue) individuals relative to total number of pilgrims (*n* = 185). (**B**) Represents the numbers (*n*) and percentages (%) of single infection (orange) and coinfection (brown) cases among infected pilgrims (*n* = 126). (**C**) The numbers (*n*) and percentages (%) of two (green), triple (purple), or quadrable (yellow) viruses relative to the number of co-infection cases (*n* = 33).

**Figure 3 pathogens-10-00315-f003:**
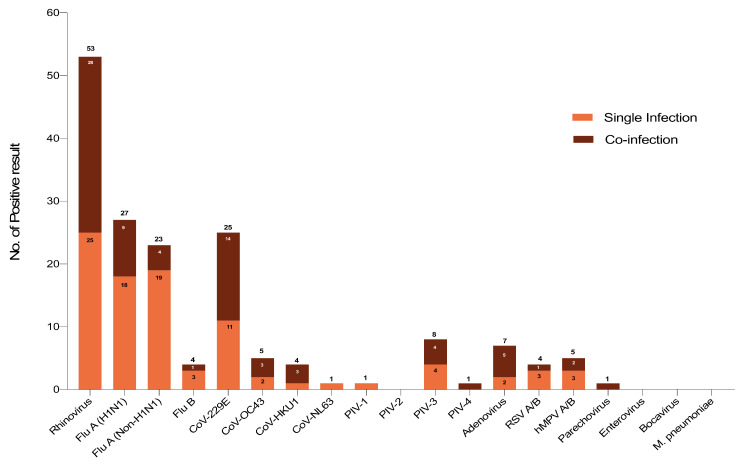
Pattern of respiratory viruses among pilgrims during the 2019 Hajj season. The virus name and the number of single or coinfection cases are shown.

**Figure 4 pathogens-10-00315-f004:**
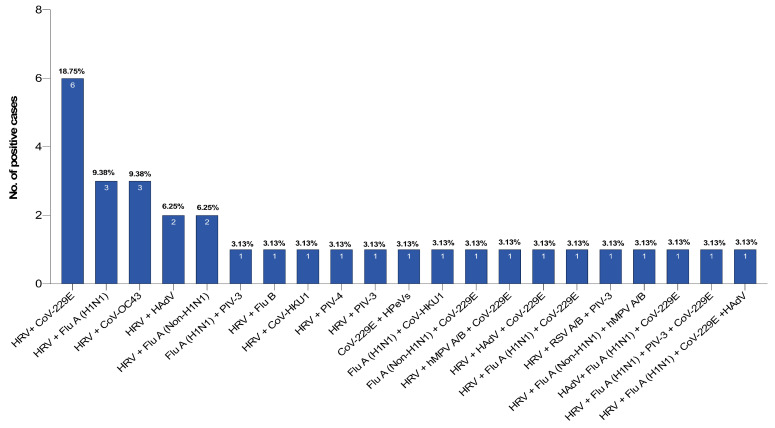
Pattern of coinfection cases among pilgrims during the 2019 Hajj season, the most common coinfections were due to the HRV with CoV-229E (18.75%), then with influenza A (H1N1) or CoV-OC43 (9.38%), with influenza A (non-H1N1) or human adenovirus (6.25%) followed by others viral combinations (3.13%).

**Table 1 pathogens-10-00315-t001:** Demographic features of the patients with acute respiratory tract infections (RTIs) at the Hajj 2019.

Characteristics	Description	*n* (%) *	Infected *n* (%) **	*p* Value ***
Gender	Male	122 (65.95%)	81 (64.29%)	0.486
	Female	63 (34.05%)	45 (35.71%)	
Age group	<60	71 (39.46%)	42 (33.33%)	0.039
	≥60	114 (60.54%)	84 (66.67%)	
Total		185	126 (68.11%)	

* Percentages (%) are relative to the total number of pilgrims (*n* = 185). ** Percentages (%) are relative to the total number of infected pilgrims (*n* = 126). *** *p* values were calculated using a chi-squared test with *p* ≤ 0.05 considered as statistically significant.

**Table 2 pathogens-10-00315-t002:** Demography of the patients with acute RTIs included in the study in Hajj 2019.

Infecting Virus	Total	Gender		Age Groups	
		Male	Female	<60	≥60
Rhinovirus	53	38 (71.70%)	15 (28.30%)	14 (26.42%)	39 (73.58%)
Influenza A (H1N1)	27	14 (51.85%)	13 (48.15%)	7 (25.93%)	20 (74.07%)
Influenza A (Non-H1N1)	23	16 (69.57%)	7 (30.43%)	12 (52.17%)	11 (47.83%)
Influenza B	4	1 (25.00%)	3 (75.00%)	3 (75.00%)	1 (25.00%)
CoV-229E	25	14 (56.00%)	11 (44.00%)	9 (36.00%)	16 (64.00%)
CoV-OC43	5	2 (40.00%)	3 (60.00%)	0 (0.00%)	5 (100.00%)
CoV-HKU1	4	3 (75.00%)	1 (25.00%)	2 (50.00%)	2 (50.00%)
CoV-NL63	1	0 (0.00%)	1 (100.00%)	0 (0.00%)	1 (100.00%)
PIV-1	1	1 (100.00%)	0 (0.00%)	0 (0.00%)	1 (100.00%)
PIV-2	0	0 (0.00%)	0 (0.00%)	0 (0.00%)	0 (0.00%)
PIV-3	8	8 (100.00%)	0 (0.00%)	2 (25.00%)	6 (75.00%)
PIV-4	1	1 (100.00%)	0 (0.00%)	0 (0.00%)	1 (100.00%)
Adenovirus	7	5 (71.43%)	2 (28.57%)	1 (14.29%)	6 (85.71%)
RSV A/B	4	4 (100.00%)	0 (0.00%)	2 (50.00%)	2 (50.00%)
HMPV A/B	5	4 (80.00%)	1 (20.00%)	2 (40.00%)	3 (60.00%)
Parechovirus	1	0 (0.00%)	1 (100.00%)	0 (0.00%)	1 (100.00%)
Enterovirus	0	0 (0.00%)	0 (0.00%)	0 (0.00%)	0 (0.00%)
Bocavirus	0	0 (0.00%)	0 (0.00%)	0 (0.00%)	0 (0.00%)
*M. pneumoniae*	0	0 (0.00%)	0 (0.00%)	0 (0.00%)	0 (0.00%)

## Data Availability

All data related to the study are available within this manuscript.

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
