# Peer review of "Pattern of Respiratory Viruses among Pilgrims during 2019 Hajj Season Who Sought Healthcare Due to Severe Respiratory Symptoms"

_pathogens, 2021, doi:10.3390/pathogens10030315_

Round 1

Reviewer 1 Report

  1. Please describe the primers, probes, conditions and detection limits in detail for Real-Time PCR in the 2.3 subsection of the “Materials and Methods” section.
  2. Please indicate the significance of Figure 2 in the “Discussion” section.
  3. All figures including Figure 1, 2, 3, and 4 are fuzzy, please replace them with clear ones.

Author Response

RESPONSES TO REVIEWER 1:

  1. Please describe the primers, probes, conditions, and detection limits in detail for Real-Time PCR in the 2.3 subsection of the “Materials and Methods” section.

RESPONSE: Please note that we used a commercially available multiplex PCR kit in this study, and the assays were performed according to the manufacturer's instruction with no modifications. We have given details of the manufacturer and catalogue number.  Specific details of primers, probes and detection limits are copyrighted. Several papers have validated the use of this kits against in-house developed primers/probes (Malhotra et al 2016; Sakthivel et al 2012; Salez et al 2015).

  1. Please indicate the significance of Figure 2 in the “Discussion” section.

RESPONSE: We thank the reviewer for this valuable comment. We have now indicated the significance of Figure 2 in the “Discussion” section as follows:

"As per pilgrims' home country, most cases associated with those arrived from Indonesia, India and Egypt (10 or more cases) (Figure 1). Cases of single and co-infection with up to 4 viruses have been identified (Figure 2). Indeed, out of the 126 pilgrims with PCR positive, single infection was detected in 93 (74%) patients. Co-infection was detected in the remaining 33 (26%) cases as follows: 25 patients with two viruses, 6 patients with three viruses, and 2 patients with 4 viruses (Figure 2)"

  1. All figures including Figure 1, 2, 3, and 4 are fuzzy, please replace them with clear ones.

RESPONSE:  Our apologies:  We have replaced all of them with high resolution figures.

Reviewer 2 Report

Review Pathogens 10999580

Alsayed et al. describe the frequency of viral single and co-infections during a five day study at a single hospital. The study is intersting, well written, the data clearly presented and well discussion.

Importantly, all fugures need to be submitted as high-resolution images, since they were hard to read in the mansucript.

Specific questions tob e adressed:  

By which selection process have been the patients been included into the study? Please add.

What kind of respiratory tract infection infections have the patients had? Please add.

Were the patients hospitalized or only ambulatory patients? Please add.

How many patients presented in the emergency room during the 5-day study in total? Can the authors give an estimation , how many RTIs to expect in that specifical hospital during Hajj pilgrimage? Please add.

Author Response

RESPONSES TO REVIEWER 2:

  1. Alsayed et al. describe the frequency of viral single and co-infections during a five-day study at a single hospital. The study is interesting, well written, the data clearly presented and well discussion.

RESPONSE: We thank to the reviewer for kind supportive comments

  1. Importantly, all figures need to be submitted as high-resolution images, since they were hard to read in the manuscript.

RESPONSE: Our apologies. We have replaced all of them with high resolution figures.

  1. Specific questions to be addressed:
  2. By which selection process have been the patients been included into the study? Please add.

RESPONSE: Thank you. We have clarified as follows under Study design as follows:

This was a cross-sectional analytical study which screened for respiratory viruses in the nasopharyngeal swabs of pilgrims with respiratory tract illnesses presenting to healthcare facilities in the holy places during the 2019 Hajj.

Study population and enrolment sites: Pilgrims with severe acute respiratory tract symptoms who presented to seven healthcare facilities in Makkah, Arafat, and Mina during the first five days of the 2019 Hajj.

Clinical Specimens and patient information: Nasopharyngeal swabs (NS) were collected in viral transport media (VTM) and stored at -80oC in the Special Infectious Agents Unit (SIAU), King Fahd Medical Research Center (KFMRC). Demographic data including nationality, age, and gender from all participants was also obtained.

  1. What kind of respiratory tract infection infections have the patients had? Please add.

RESPONSE: All patients had severe acute respiratory symptoms on presentation to healthcare facilities. Figure 2 and figure 3 give specific details of viruses detected in nasopharyngeal swabs

  1. Were the patients hospitalized or only ambulatory patients? Please add.

RESPONSE: Thank you. We have now added this sentence in the study design: “All patients were hospitalized and initially were tested for MERS-CoV. All patients enrolled tested negative for MERS-CoV.”

  1. How many patients presented in the emergency room during the 5-day study in total? Can the authors give an estimation, how many RTIs to expect in that specifical hospital during Hajj pilgrimage? Please add.

RESPONSE: Thank you. There are over 14 major medical facilities in addition to multiple primary care centers in the holy places that serve more than 2 and half million pilgrims during Hajj. Unfortunately, information about the number of patients presented in the emergency room during the 5-day study in total is not available to us.

Round 2

Reviewer 1 Report

No.

Author Response

We thank the reviewer for his feedback.

Reviewer 2 Report

Alsayed  et al. followed most of the reviewer request adequatly. However, the question what kind of respiratory tract infection infections the patients had, remains unresolved. Already in the title of the paper the authors state, that all patients were hospitalized for severe respiratory symptoms. But what were these symptoms (have all patients had a (radiologically confirmed pneumonia; oxygen demand etc.))?

Author Response

Comment: Alsayed  et al. followed most of the reviewer request adequatly. However, the question what kind of respiratory tract infection infections the patients had, remains unresolved. Already in the title of the paper the authors state, that all patients were hospitalized for severe respiratory symptoms. But what were these symptoms (have all patients had a (radiologically confirmed pneumonia; oxygen demand etc.))?

Response: Thank you for the comment. The symptoms and diagnosis were observed by the clinician as acute respiratory infections but we do not have the data for these symptoms nor their radiology graphs.